# Cohesin mutations are synthetic lethal with stimulation of WNT signaling

Chue Vin Chin[1,2,3], Jisha Antony[1,2,3], Sarada Ketharnathan[1,3], Anastasia Labudina[1,3], Gregory Gimenez[1,3], Kate M Parsons[4], Jinshu He[4], Amee J George[4], Maria Michela Pallotta[5], Antonio Musio[5], Antony Braithwaite[1,2], Parry Guilford[6], Ross D Hannan[4,7,8,9], Julia A Horsfield[1,2,3]*

[1]Department of Pathology, Otago Medical School, University of Otago, Dunedin, New Zealand; [2]Maurice Wilkins Centre for Molecular Biodiscovery, The University of Auckland, Auckland, New Zealand; [3]Genetics Otago Research Centre, University of Otago, Dunedin, New Zealand; [4]The John Curtin School of Medical Research, The Australian National University, Canberra, Australia; [5]Istituto di Ricerca Genetica e Biomedica (IRGB), Consiglio Nazionale delle Ricerche (CNR), Pisa, Italy; [6]Department of Biochemistry, University of Otago, Dunedin, New Zealand; [7]Department of Biochemistry and Molecular Biology, University of Melbourne, Parkville, Australia; [8]Sir Peter MacCallum Department of Oncology, University of Melbourne, Parkville, Australia; [9]School of Biomedical Sciences, University of Queensland, St Lucia, Australia

*For correspondence:
julia.horsfield@otago.ac.nz

Competing interests: The authors declare that no competing interests exist.

**Abstract** Mutations in genes encoding subunits of the cohesin complex are common in several cancers, but may also expose druggable vulnerabilities. We generated isogenic MCF10A cell lines with deletion mutations of genes encoding cohesin subunits SMC3, RAD21, and STAG2 and screened for synthetic lethality with 3009 FDA-approved compounds. The screen identified several compounds that interfere with transcription, DNA damage repair and the cell cycle. Unexpectedly, one of the top 'hits' was a GSK3 inhibitor, an agonist of Wnt signaling. We show that sensitivity to GSK3 inhibition is likely due to stabilization of β-catenin in cohesin-mutant cells, and that Wnt-responsive gene expression is highly sensitized in *STAG2*-mutant CMK leukemia cells. Moreover, Wnt activity is enhanced in zebrafish mutant for cohesin subunits *stag2b* and *rad21*. Our results suggest that cohesin mutations could progress oncogenesis by enhancing Wnt signaling, and that targeting the Wnt pathway may represent a novel therapeutic strategy for cohesin-mutant cancers.

## Introduction

The cohesin complex is essential for sister chromatid cohesion, DNA replication, DNA repair, and genome organization. Three subunits, SMC1A, SMC3, and RAD21, form the core ring-shaped structure of human cohesin (*Dorsett and Ström, 2012*; *Horsfield et al., 2012*). A fourth subunit of either STAG1 or STAG2 binds to cohesin by contacting RAD21 and SMC subunits (*Shi et al., 2020*), and is required for the association of cohesin with DNA (*Dorsett and Ström, 2012*; *Horsfield et al., 2012*; *Shi et al., 2020*). The STAG subunits of cohesin are also capable of binding RNA in the nucleus (*Pan et al., 2020*). Cohesin associates with DNA by interaction with loading factors NIPBL and MAU2 (*Wendt, 2017*), its stability on DNA is regulated by the acetyltransferases ESCO1 (*Wutz et al., 2020*) and ESCO2 (*van der Lelij et al., 2009*), and its removal is facilitated by PDS5 and WAPL (*Wutz et al., 2017*; *Shintomi and Hirano, 2009*). The cohesin ring itself acts as a molecular motor to extrude DNA loops, and this activity is thought to underlie its ability to organize the genome (*Vian et al., 2018*; *Mayerova et al., 2020*). Cohesin works together with CCCTC-binding

factor (CTCF) to mediate three-dimensional genome structures, including enhancer-promoter loops that instruct gene accessibility and expression (*Hansen, 2020*; *Rowley and Corces, 2018*). The consequences of cohesin mutation therefore manifest as chromosome segregation errors, DNA damage, and deficiencies in genome organization leading to gene expression changes.

All cohesin subunits are essential to life: homozygous mutations in genes encoding complex members are embryonic lethal (*Horsfield et al., 2012*). However, haploinsufficient germline mutations in *NIPBL*, *ESCO2*, and in cohesin genes, cause human developmental syndromes known as the 'cohesinopathies' (*Horsfield et al., 2012*). Somatic mutations in cohesin genes are prevalent in several different types of cancer, including bladder cancer (15–40%), endometrial cancer (19%), glioblastoma (7%), Ewing's sarcoma (16–22%) and myeloid leukemias (5–53%) (*De Koninck and Losada, 2016*; *Hill et al., 2016*; *Waldman, 2020*). The prevalence of cohesin gene mutations in myeloid malignancies (*Kon et al., 2013*; *Papaemmanuil et al., 2016*; *Thol et al., 2014*; *Thota et al., 2014*; *Yoshida et al., 2013*) reflects cohesin's role in determining lineage identity and differentiation of hematopoietic cells (*Galeev et al., 2016*; *Mazumdar et al., 2015*; *Mullenders et al., 2015*; *Viny et al., 2015*). Of the cohesin genes, *STAG2* is the most frequently mutated, with about half of cohesin mutations in cancer involving *STAG2* (*Waldman, 2020*).

While cancer-associated mutations in genes encoding RAD21, SMC3, and STAG1 are always heterozygous (*Thota et al., 2014*; *Kon et al., 2013*; *Tsai et al., 2017*), mutations in the X chromosome-located genes *SMC1A* and *STAG2* can result in complete loss of function due to hemizygosity (males), or silencing of the wild type during X-inactivation (females). STAG2 and STAG1 have redundant roles in cell division, therefore complete loss of STAG2 is tolerated due to partial compensation by STAG1. Loss of both STAG2 and STAG1 leads to lethality (*Benedetti et al., 2017*; *van der Lelij et al., 2017*). STAG1 inhibition in cancer cells with STAG2 mutation causes chromosome segregation defects and subsequent lethality (*Liu et al., 2018*). Therefore, although partial depletion of cohesin can confer a selective advantage to cancer cells, a complete block of cohesin function will cause cell death. The multiple roles of cohesin provide an opportunity to inhibit the growth of cohesin-mutant cancer cells via chemical interference with pathways that depend on normal cohesin function. For example, poly ADP-ribose polymerase (PARP) inhibitors were previously shown to exhibit synthetic lethality with cohesin mutations (*Waldman, 2020*; *Liu et al., 2018*; *Mondal et al., 2019*; *McLellan et al., 2012*; *O'Neil et al., 2013*). PARP inhibitors prevent DNA double-strand break repair (*Zaremba and Curtin, 2007*), a process that also relies on cohesin function.

To date, only a limited number of compounds have been identified as inhibitors of cohesin-mutant cells (*Waldman, 2020*). Here, we sought to identify additional compounds of interest by screening libraries of FDA-approved molecules against isogenic MCF10A cells with deficiencies in RAD21, SMC3, or STAG2. Unexpectedly, our screen identified a novel sensitivity of cohesin-deficient cells to a GSK3 inhibitor that acts as an agonist of the Wnt signaling pathway. We found that β-catenin stabilization upon cohesin deficiency likely contributes to an acute sensitivity of Wnt target genes. The results raise the possibility that sensitization to Wnt signaling in cohesin-mutant cells may participate in oncogenesis, and suggest that Wnt agonism could be therapeutically useful for cohesin-mutant cancers.

## Results

### Cohesin gene deletion in MCF10A cells results in minor cell cycle defects

To avoid any complications with pre-existing oncogenic mutations, we chose the relatively 'normal' MCF10A line for creation and screening of isogenic deletion clones of cohesin genes *SMC3*, *RAD21*, and *STAG2*. MCF10A is a near-diploid, immortalized, breast epithelial cell line that exhibits normal epithelial characteristics (*Tait et al., 1990*) and has been successfully used for siRNA and small molecule screening (*Telford et al., 2015*). Two sgRNAs per gene were designed targeting the 5' and 3' UTR regions, respectively, of *RAD21*, *SMC3*, and *STAG2* genes. Single cells were isolated and grown into clones that were genotyped for complete gene deletions (*Figure 1*, *Supplementary file 1*). We isolated several *RAD21* and *SMC3* deletion clones, and selected single clones for further characterization that grew normally and were essentially heterozygous. In the selected *RAD21* deletion clone, one of three *RAD21* alleles (on chromosome 8, triploid in MCF10A) was confirmed deleted, with one

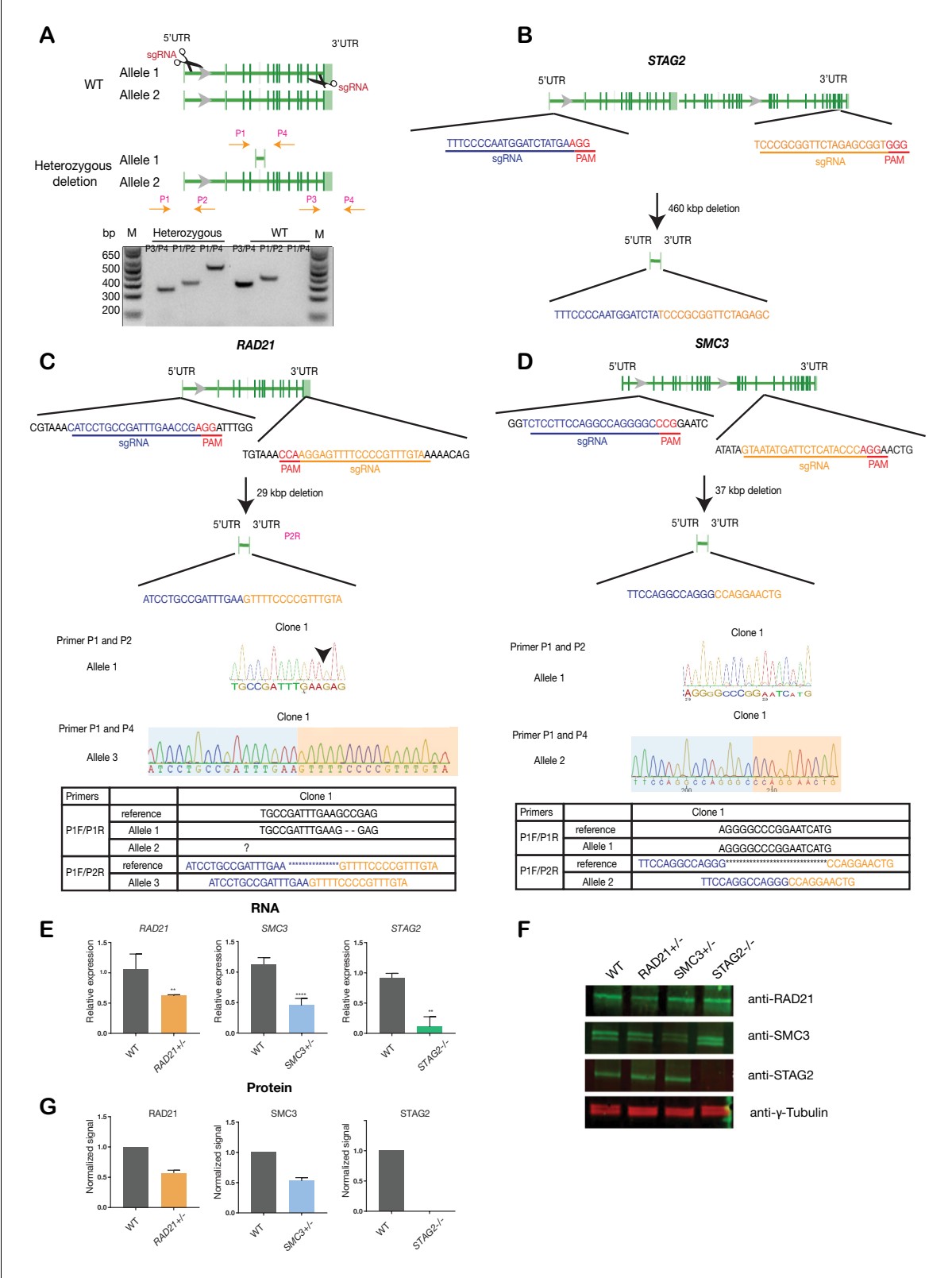

**Figure 1.** Creation of MCF10A isogenic cell lines with cohesin gene deletions. (**A**) Top, schematic diagram shows the deletion strategy for genes encoding cohesin subunits RAD21, SMC3, and STAG2 using two sgRNAs targeting the 5'UTR and the 3'UTR of each gene. Bottom, heterozygous clones were identified by PCR using specific primer pairs flanking the deletion region. Representative DNA gel shows the PCR products yielded using specific primer pairs for MCF10A parental and *RAD21+/-* deletion clone. M, ladder marker. (**B,C,D**) Schematic deletion strategy and summary of the

*Figure 1 continued on next page*

*Figure 1 continued*

allele sequences for the *STAG2* homozygous deletion clone, and the *RAD21* and *SMC3* heterozygous deletion clones. (E) RNA levels of the targeted genes in MCF10A cohesin-deficient clones. (F) Representative immunoblot and (G) quantification of cohesin protein levels. γ-tubulin was used as loading control. n = 3 independent experiments, mean ±s.d., one-tailed student *t* test: **p≤0.01; ****p≤0.0001. Guide RNAs and PCR primers can be found in *Supplementary file 1*.

wild type and one undetermined allele. In the selected *SMC3* deletion clone, one of two *SMC3* alleles (on chromosome 10) was deleted. In the selected *STAG2* deletion clone, homozygous loss of *STAG2* was determined by the absence of STAG2 mRNA and protein. For convenience here, we named the three cohesin-mutant clones *RAD21+/-*, *SMC3+/-*, and *STAG2-/-*.

The *RAD21+/-*, *SMC3+/-*, and *STAG2-/-* clones had essentially normal cell cycle progression when compared to parental cells, although the *RAD21+/-* and *SMC3+/-* clones proliferated more slowly than the others (*Figure 2A,B*). Chromosome spreads revealed that only the *STAG2-/-* clone had noteworthy chromosome cohesion defects characterized by partial or complete loss of chromosome cohesion and gain or loss of more than one chromosome (*Figure 2C,D*; *Figure 2—figure supplement 1A*). Remarkably, *SMC3+/-* cells had noticeably larger nuclei that appeared to be less dense (*Figure 2E,F*; *Figure 2—figure supplement 1B*), possibly owing to decompaction of DNA in this clone. *RAD21+/-* and *SMC3+/-* clones exhibited occasional lagging chromosomes and micronuclei, while the *STAG2-/-* clone did not (*Figure 2G*; *Figure 2—figure supplement 1C–E*). Cell growth and morphology in monolayer culture was essentially normal in all three cohesin-mutant clones (*Figure 2—figure supplement 2*).

Overall, the cohesin deletion clones appear to have infrequent but specific cell cycle anomalies that are shared between some, but not all clones. Anomalies include loss of chromosome cohesion, lagging chromosomes, or micronuclei, but these features do not appear to majorly impact on cell cycle progression or morphology.

## Cohesin-depleted cells have altered nucleolar morphology and are sensitive to DNA damaging agents

Cohesin-deficient cells have been demonstrated to display compromised nucleolar morphology and ribosome biogenesis (*Bose et al., 2012*; *Harris et al., 2014*), as well as sensitivity to DNA damaging agents (*Bailey et al., 2014*; *Mondal et al., 2019*). Analysis of our *RAD21+/-*, *SMC3+/-*, and *STAG2-/-* clones confirmed these findings. Cohesin-deleted cells in steady-state growth had abnormal nucleolar morphology, as revealed by fibrillarin and nucleolin staining (*Figure 3—figure supplement 1*). Furthermore, we found that treatment with DNA intercalator/transcription inhibitor Actinomycin D caused marked fragmentation of nucleoli in all three cohesin-deficient cell lines as determined by fibrillarin staining (*Figure 3A,B*). Actinomycin D treatment increased γ-H2AX and TP53 in the nuclei of *RAD21+/-* and *SMC3+/-* cells, in particular, indicating that these cells are compromised for DNA damage repair relative to the parental MCF10A cells (*Figure 3C–F*). In contrast, immunostaining for γ-H2AX and TP53 levels were comparable at baseline between cohesin-deficient clones and parental cells. Interestingly, the *STAG2-/-* deletion clone was much more resistant to DNA damage caused by Actinomycin D (*Figure 3C–F*), even though nucleoli are abnormal in this clone (*Figure 3—figure supplement 1*).

Overall, increased susceptibility of *RAD21+/-* and *SMC3+/-* clones to DNA damage is consistent with cohesin's role in DNA double-strand break repair (*Sjögren and Ström, 2010*), and highlights the different requirements for RAD21 and SMC3 versus STAG2 in this process. Cohesin mutations were previously shown to sensitize cells to PARP inhibitor, Olaparib (*Mondal et al., 2019*; *Matto et al., 2015*). We confirmed a mild to moderate sensitivity to Olaparib in our cohesin-deficient MCF10A clones relative to parental cells (*Figure 3—figure supplement 2*).

Collectively, characterization of our cohesin-deficient clones provided confidence that they represent suitable models for synthetic lethal screening. To confirm that the phenotypes of our chosen clones are representative, we selected a further two clones with heterozygous deletions in *SMC3* and *RAD21*, and monitored their growth, morphology and chromosome cohesion (*Figure 3—figure supplement 3*). These analyses showed that the two additional deletion clones were similar to those already characterized (*Figures 1–3*), providing surety that our cohesin-deficient cell lines have properly representative phenotypes. Furthermore, none of the cohesin-deficient clones exhibited

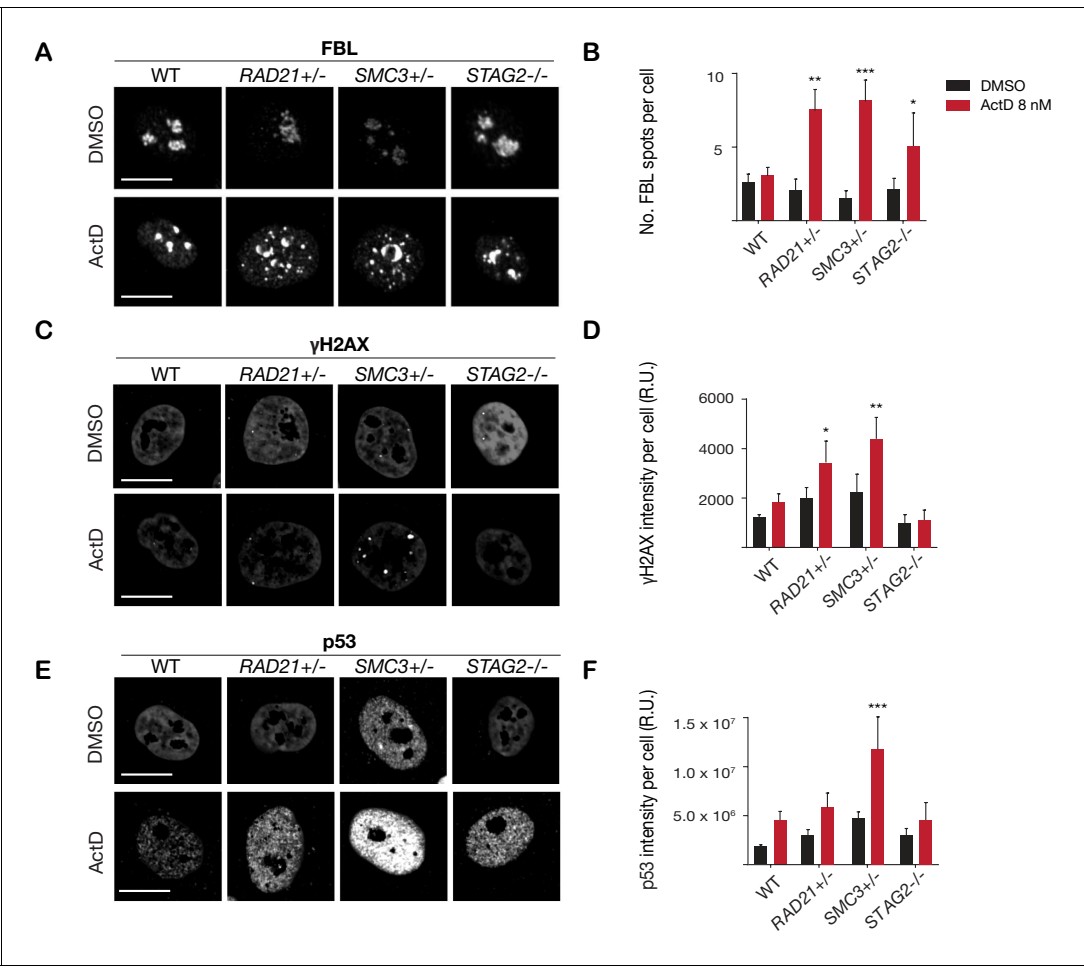

**Figure 3.** Cohesin-deficient cells have increased sensitivity to nucleolar stress and DNA damaging agents. (**A**) Representative image and (**B**) quantification of nucleolar dispersal observed in parental (WT) MCF10A cells and cohesin-deficient clones after exposure to a DNA damaging agent, Actinomycin D (ActD) 8 nM. Fibrillarin (FBL) staining was used as a marker for nucleoli. (**C**) Representative image and (**D**) quantification of DNA damage foci observed in parental (WT) MCF10A cells and cohesin-deficient clones after exposure to ActD. An antibody detecting γH2AX was used to visualize foci of DNA double-strand breaks. (**E**) Representative image and (**F**) quantification of nuclear p53 in parental (WT) MCF10A cells and cohesin-deficient clones after exposure to ActD. A minimum of 500 cells was examined per individual experiment. n = 3 independent experiments, mean ± s.d., one-way ANOVA: *p≤0.05; **p≤0.01; ***p≤0.0005. Scale bar, 15 µM. Source data is available for **Figure 2B,D,F** in **Figure 3—source data 1**.

The online version of this article includes the following source data and figure supplement(s) for figure 3:

**Source data 1.** Raw data for **Figure 3**.
**Figure supplement 1.** Cohesin-deficient cells show altered nucleolar morphology.
**Figure supplement 2.** PARP sensitivity of cohesin-deficient cells.
**Figure supplement 3.** Data replication with additional MCF10A isogenic cell lines with cohesin gene deletions.

enhanced cell death compared with parental MCF10A cells (*Figure 2B*; *Figure 3—figure supplement 3*).

## A synthetic lethal screen of FDA-approved compounds identifies common sensitivity of cohesin-mutant cells to WNT activation and BET inhibition

To identify additional compounds that inhibit the growth of cohesin-deficient cells, we screened the cohesin-deficient MCF10A cell lines with five dose concentrations (1–5000 nM) of 3009 compounds, including FDA-approved compounds (2399/3009), kinase inhibitors (429/3009), and epigenetic modifiers (181/3009) (*Figure 4A*). DMSO and Camptothecin were included as negative and positive viability controls, respectively (*Figure 4—figure supplement 1*). We assayed cell viability after 48 hr of compound treatment.

Synthetic lethal candidate compounds were ranked based on the differential area over curve (AOC) values that are derived from a growth rate-based metric (GR) (*Figure 4B–D*; *Figure 4—source data 1*). The GR metric takes into account the varying growth rate of dividing cells to mitigate inconsistent comparison of compound effects across cohesin-deficient cell lines (*Hafner et al., 2016*). Compounds that caused ≥30% growth inhibition in cohesin-deficient clones compared with parental MCF10A cells were selected for further analysis. The screen identified candidate 206 synthetic lethal compounds, of which 18 inhibited all three cohesin-deficient cell lines ≥ 30% more than the parental MCF10A cells (*Figure 4E*; *Figure 4—source data 2*; *Table 1*; *Table 1—source data 1*).

Most of the 206 compounds inhibited at least two cohesin-deficient cell lines and were classed in similar categories of inhibitor (*Figure 4—figure supplement 2A–D*; *Figure 4—source data 2*; *Table 1*). Of the 206 primary screen hits, 85 (including the 18 that inhibited all three cohesin-deficient clones, plus the most effective candidates from each inhibitor category) were subjected to secondary screening in an 11-point dose curve ranging from 0.5 nM to 10 µM. Notable sensitive pathways included: DNA damage repair, the PI3K/AKT/mTOR pathway, epigenetic control of transcription, and stimulation of the Wnt signaling pathway (*Table 1*; *Table 1—source data 2*; *Figure 4—source data 2*).

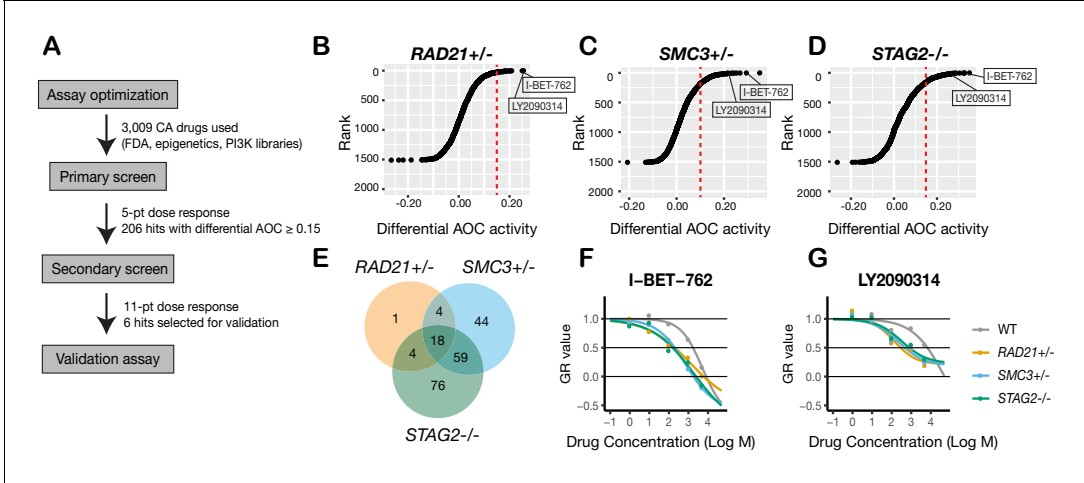

**Figure 4.** A synthetic lethal screen identifies common sensitivity of cohesin-deficient cells to WNT activation and BET inhibition. (**A**) Schematic overview of the synthetic lethal screen. (**B,C,D**) Overview of the differential area over the curve (AOC) activity of all compounds tested in cohesin-deficient cell lines relative to parental MCF10A cells in the primary screen. A threshold of differential AOC ≥ 0.15 (red dashed lines) was used to filter candidate compounds of interest. (**E**) Venn diagram showing the number of common and unique compounds that inhibited *RAD21+/-*, *SMC3+/-*, and *STAG2-/-* in the primary screen. (**F,G**) Dose-response curves of I-BET-762 and LY2090314. Source data is available for B–E in *Figure 4—source data 1*.

The online version of this article includes the following source data and figure supplement(s) for figure 4:

**Source data 1.** Raw cell counts for SL screen compound treatments.
**Source data 2.** Table and AOC measurements of all hit compounds from the screen.
**Figure supplement 1.** Synthetic lethal screen controls.
**Figure supplement 2.** Categories of compounds that differentially inhibit cohesin-deficient cells.

**Table 1.** Significant inhibitors of all three cohesin-deficient clones.

| Compound | Rank 1° screen | Rank 2° screen | Target | Pathway | 1° differential AOC activity | | |
| --- | --- | --- | --- | --- | --- | --- | --- |
| | | | | | RAD21+/- | SMC3+/- | STAG2-/- |
| WAY-600 | 1 | 3 | mTORC1/2 | PI3K/AKT/mTOR | 0.30 | 0.37 | 0.32 |
| I-BET-762 | 2 | 8 | BET proteins | Epigenetics | 0.25 | 0.27 | 0.29 |
| LY2090314 | 3 | 6 | GSK3 | WNT | 0.25 | 0.22 | 0.25 |
| Vistusertib (AZD2014) | 4 | 1 | mTORC1/2 | PI3K/AKT/mTOR | 0.18 | 0.33 | 0.26 |
| P276-00 | 5 | 17 | CDK1/4/9 | Cell Cycle | 0.17 | 0.29 | 0.28 |
| MK-8745 | 6 | 22 | Aurora A | Cell Cycle | 0.18 | 0.30 | 0.26 |
| Ethacridine lactate | 7 | 28 | Anti-infection | Microbiology | 0.18 | 0.25 | 0.27 |
| CUDC-101 | 8 | 36 | EGFR, HDAC, HER2 | Epigenetics | 0.16 | 0.25 | 0.24 |
| Dabrafenib (GSK2118436) | 9 | 9 | BRAF | MAPK | 0.18 | 0.27 | 0.18 |
| SAR131675 | 10 | 34 | VEGFR3 | Protein Tyrosine Kinase | 0.15 | 0.17 | 0.33 |
| ZM 447439 | 11 | 41 | Aurora A/B | Cell Cycle | 0.16 | 0.22 | 0.24 |
| Gitoxigenin Diacetate | 12 | 32 | NA | Other | 0.17 | 0.24 | 0.20 |
| UNC669 | 13 | 62 | Epigenetic Reader Domain | Epigenetics | 0.18 | 0.23 | 0.20 |
| 4-Phenylbutyric Acid | 14 | 29 | Endoplasmic reticulum stress | Other | 0.20 | 0.17 | 0.20 |
| Ipatasertib (GDC-0068) | 15 | 12 | AKT | PI3K/AKT/mTOR | 0.21 | 0.19 | 0.16 |
| VX-702 | 16 | 43 | P38 MAPK | MAPK | 0.15 | 0.19 | 0.21 |
| RVX-208 | 17 | 58 | BET proteins | Epigenetics | 0.16 | 0.20 | 0.17 |
| Dihydroergotamine mesylate | 18 | 49 | NA | Other | 0.16 | 0.19 | 0.16 |
| Olaparib | 351 | | PARP1/2 | DNA Damage | 0.11 | 0.13 | 0.22 |

The online version of this article includes the following source data for Table 1:

Source data 1. Compounds with growth inhibitory activity (AOC) ranked to produce *Table 1*.

Source data 2. Compounds effective in the secondary screen ranked to produce *Table 1*.

The identification of DNA damage repair inhibitors in our screen agrees with previous studies showing synthetic lethality of PARP inhibition with cohesin mutation (*McLellan et al., 2012*). Differential sensitivity of the cohesin deletion cell lines to PI3K/AKT/mTOR inhibitors is consistent with the observed nucleolar deficiencies in cohesin-mutant cells (*Figure 3—figure supplement 1*). The PI3K/AKT/mTOR pathway stimulates ribosome biogenesis and rDNA transcription; because rDNA is contained in nucleoli, it is likely that rDNA transcription and ribosome production is already compromised in the cohesin gene deletion cell lines. We had previously shown that BET inhibition is effective in blocking precocious gene expression in the chronic myelogenous leukemia cell line K562 containing a *STAG2* mutation (*Antony et al., 2020*). Growth inhibition of cohesin-deficient MCF10A cells by I-BET-762 (*Figure 4F*; *Figure 4—figure supplement 2E*) reinforces the idea that targeting BET could be therapeutically effective in cohesin-mutant cancers.

Interestingly, we found that LY2090314, a GSK3 inhibitor and stimulator of the Wnt signaling pathway, inhibits all three cohesin deletion lines (*Table 1*, *Figure 4G*, *Figure 4—figure supplement 2F*). LY2090314 also inhibited the growth of K562 STAG2 R614* mutant leukemia cells that we had previously characterized (*Antony et al., 2020*; *Figure 4—figure supplement 2G*) as well as the two additional MCF10A *RAD21*- and *SMC3*-deficient clones (*Figure 3—figure supplement 3G*). Wnt signaling appears to act upstream of cohesin (*Xu et al., 2014*; *Estarás et al., 2015*), and also to be primarily downregulated downstream of cohesin mutation (*Mills et al., 2018*; *Schuster et al., 2015*;

*Avagliano et al., 2017*). Therefore, we were prompted to further investigate why Wnt stimulation via GSK3 inhibition might cause lethality in cohesin-deficient cells.

## Stabilization of β-catenin in cohesin-deficient MCF10A cells

In 'off' state of canonical Wnt signaling, β-catenin forms a complex including Axin, APC, GSK3, and CK1 proteins. β-catenin phosphorylated by CK1 and GSK3 is recognized by the E3 ubiquitin ligase β-Trcp, which targets β-catenin for proteasomal degradation. Activation of Wnt signaling by ligand binding, or by GSK3 inhibition, releases β-catenin, allowing it to accumulate in the nucleus where it binds TCF to activate Wnt target gene transcription (*Logan and Nusse, 2004*; *Clevers et al., 2014*). We found that there was no difference in GSK3 levels between parental MCF10A cells and the cohesin-deficient clones, and upon treatment of cells with LY2090314, levels of GSK3 were reduced in all cells as expected (*Figure 5—figure supplement 1*). In contrast, in untreated cells we found that β-catenin is stabilized in all three cohesin-deficient clones compared with parental MCF10A cells. When Wnt signaling is in the 'off' state, phospho-Ser33/37/Thr41 marks β-catenin targeted for degradation. Membrane-associated phospho-Ser33/37/Thr41-β-catenin was increased in cohesin-deficient clones, and disappeared upon treatment with LY2090314 (*Figure 5A*). Phospho-Ser675 β-catenin, the active form of β-catenin that is induced upon Wnt signaling, was noticeably increased in the cytoplasm of cohesin-deficient cells following treatment with LY2090314 (*Figure 5B,C*). The results imply that inactive β-catenin that would normally be targeted for degradation is instead stabilized in cohesin-deficient cells, and is available to be converted into the active form following inhibition of GSK3.

To determine if stabilization of β-catenin is conserved in a second model of cohesin-mutant cancer, we performed an identical LY2090314 treatment on HCT116 cells that were stably transfected to express two *SMC1A* mutations identified in human colorectal carcinomas, c.2027A > G (leading to p.E676G change near the hinge domain) and c.2479 C > T (leading to a truncated protein, p. Q827X) (*Cucco et al., 2014*; *Sarogni et al., 2019*). The limitation is that these cells are not a model of cohesin deficiency, but rather, one in which normal cohesin function is perturbed by expression of a mutant version of SMC1A (*Sarogni et al., 2019*). We observed an increased basal level of phosphorylated β-catenin at Ser675 in cells that express either of these SMC1A mutants. Furthermore, LY2090314 treatment markedly increased total β-catenin in both the SMC1A mutants compared with HCT116 wild type controls (*Figure 5—figure supplement 2A,B*). The results indicate that abnormally high levels of active β-catenin are also present following Wnt stimulation when a fourth subunit of cohesin, SMC1A, is perturbed.

Immunofluorescence labeling of MCF10A cells showed that in LY2090314-treated cells, the stabilized active phospho-Ser675 β-catenin (and total β-catenin, *Figure 5—figure supplement 3B*) mainly locates to puncta in the cytoplasm (*Figure 5D*). In contrast, no β-catenin accumulation was observed in DMSO-treated cells (*Figure 5—figure supplement 3A*). β-catenin-containing puncta were also observed upon WNT3A treatment of cohesin-deficient clones (*Figure 5—figure supplement 3C*). This indicates that Wnt stimulation rather than a secondary effect of LY2090314 is responsible for β-catenin accumulation. We could not reliably detect an increase of β-catenin in the nucleus of cohesin-deficient MCF10A clones by immunofluorescence (*Figure 5D*), therefore we decided to use transcriptional response to determine the consequences of β-catenin stabilization for Wnt signaling.

## Cohesin-deficient leukemia cells are hypersensitive to Wnt signaling

RNA-sequencing analysis of the cohesin gene deletion clones compared with parental MCF10A cells revealed that gene expression changes strongly track with the identity of the deleted cohesin gene (*Figure 6—figure supplement 1A*). However, expressed transcripts encoding Wnt signaling pathway components did not cluster differently in cohesin-deficient clones compared with parental MCF10A cells, and cohesin mutation-based clustering remained dominant (*Figure 6—figure supplement 1B*). We reasoned that stimulation of Wnt signaling in a more responsive cell type might be necessary to determine how stabilized β-catenin in cohesin-deficient cells affects transcription.

Myeloid leukemias are frequently characterized by cohesin mutations, particularly in *STAG2*. Furthermore, activation of Wnt signaling is associated with transformation in AML (*Wang et al., 2010*; *Beghini et al., 2012*; *Kang et al., 2020*) and AML patients were identified with mutations in *AXIN1* and *APC* that lead to stabilization of β-catenin (*Erbilgin et al., 2012*). CMK is a Down Syndrome-

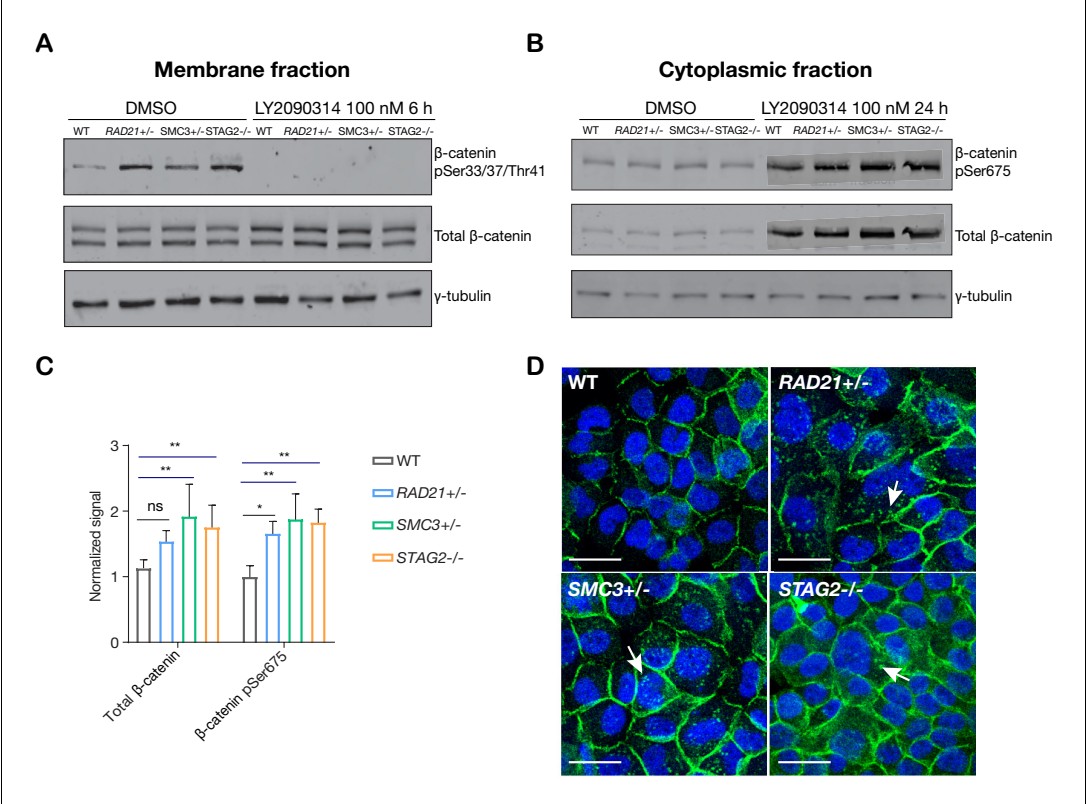

**Figure 5.** LY2090314-mediated WNT stimulation leads to β-catenin stabilization in cohesin-deficient MCF10A cells. (**A**) Immunoblot of the membrane fraction of parental (WT) and cohesin-deficient MCF10A cells shows increased basal level of β-catenin phosphorylation at Ser33/37/Thr41. (**B**) Immunoblot of the cytoplasmic fraction shows increased level of both total and phosphorylated β-catenin at Ser675 after parental (WT) and cohesin-deficient MCF10A cells were treated with LY2090314 at 100 nM for 24 hr. (**C**) Quantification of protein levels for total and phosphorylated β-catenin at Ser675. n = 3 independent experiments, mean ± s.d., one-way ANOVA: *p≤0.05, **p≤0.01. (**D**) Immunofluorescence images show cytosolic accumulation of active β-catenin in cohesin-deficient MCF10A cells treated with LY2090314 100 nM for 24 hr, relative to parental (WT) MCF10A cells. White arrows indicate puncta of β-catenin (pSer675). Scale bar = 25 μM. Full length blots and molecular size markers are available for A,B in *Figure 5—source data 1*. Source quantification data is available for C in *Figure 5—source data 2*.

The online version of this article includes the following source data and figure supplement(s) for figure 5:

**Source data 1.** Untrimmed blots for *Figure 5A, B*.
**Source data 2.** Quantitation of blots in *Figure 5A, B*.
**Source data 3.** Untrimmed blots for *Figure 5A, B*.
**Source data 4.** Untrimmed blots for *Figure 5A, B*.
**Figure supplement 1.** GSK3 levels are unaffected in cohesin-deficient MCF10A cells.
**Figure supplement 2.** LY2090314-mediated WNT stimulation leads to increased β-catenin stabilization in SMC1A mutant HCT116 cells.
**Figure supplement 3.** WNT3A phenocopies LY2090314-mediated β-catenin accumulation in the cytoplasm.

derived megakaryoblastic cell line that typifies myeloid leukemias that are particularly prone to cohesin mutation (*Yoshida et al., 2013*). We edited CMK to contain the STAG2-AML associated mutation R614* (*Antony et al., 2020*) and selected single clones with complete loss of the STAG2 protein to create the cell line CMK-*STAG2-/-* (*Figure 6—figure supplement 2A–C*).

Immunofluorescence showed that β-catenin is increased by 20% in the nuclei of CMK-*STAG2-/-* cells relative to parental cells upon WNT3A stimulation (*Figure 6A,B*). To determine immediate early transcriptional responses to Wnt signaling, RNA-sequencing was performed on CMK-*STAG2-/-* and parental CMK cells at baseline and after stimulation with WNT3A for 4 hr. Around 76% more genes were upregulated in response to WNT3A in CMK-*STAG2-/-* compared with CMK parental cells (616 in CMK-*STAG2-/-* vs 350 in CMK parental, FDR ≤ 0.05, *Figure 6C*), while around the same number were downregulated. About one quarter of differentially expressed genes overlapped between CMK-*STAG2-/-* and CMK parental cells.

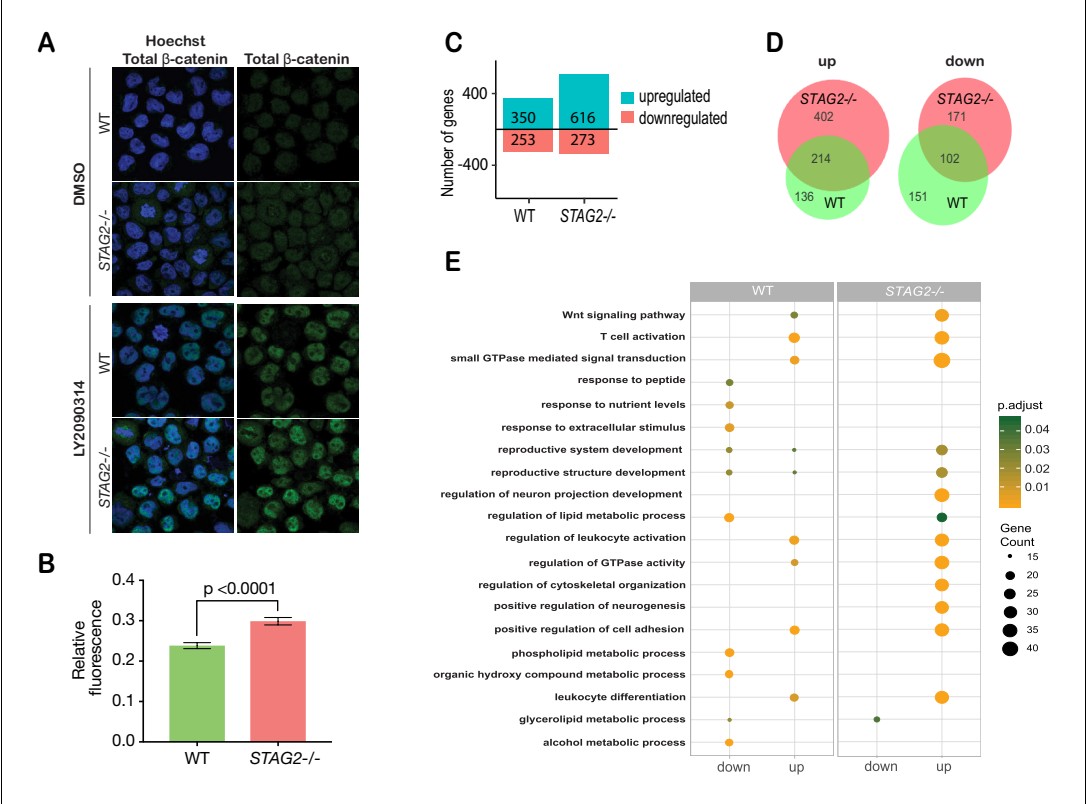

**Figure 6.** Cohesin-STAG2 mutant CMK cells show increased sensitivity to Wnt signaling. (A) Immunofluorescence images showing slightly increased nuclear accumulation of β-catenin in STAG2-CMK cells (*STAG2-/-*) compared to parental (WT) following treatment with LY2090314 at 100 nM for 24 hr. (B) Quantification of nuclear total β-catenin in parental (WT) and STAG2-CMK cells (*STAG2-/-*) cells. Fluorescence of nuclear total β-catenin was determined relative to the nuclear area. Image J was used quantify cells from 10 different confocal fields. The graph depicts s.e.m. from analyses of 170–188 cell nuclei, and the p value was calculated using a student's *t* test. (C) Histogram showing the number of genes upregulated or downregulated at FDR ≤ 0.05 upon WNT3A treatment in parental (WT) and *STAG2-/-* CMK cells. WNT3A stimulation was performed on three biological replicates, from three independent experiments. (D) Overlap of genes significantly upregulated or downregulated (FDR ≤ 0.05) upon WNT3A treatment between parental (WT) and *STAG2-/-* CMK cells. (E) Top enriched pathways (ranked by gene count) from the significantly downregulated and upregulated genes (FDR ≤ 0.05) following WNT3A treatment in *STAG2-/-* and parental (WT) CMK cells using the ClusterProfiler R package on the Gene Ontology Biological Process dataset modeling for both cell types (WT or *STAG2-/-*) and regulation pattern (up- or downregulation). Source data is available for C, D in *Figure 6—source data 1*.

The online version of this article includes the following source data and figure supplement(s) for figure 6:

**Source data 1.** Gene expression data for *Figure 6*.
**Figure supplement 1.** RNA-sequencing profiling of cohesin-deficient MCF10A cells.
**Figure supplement 2.** Enhanced sensitivity of cohesin-mutant CMK cells to Wnt stimulation.

Strikingly, 402 genes that were not Wnt-responsive in CMK parental cells became Wnt-responsive upon introduction of the *STAG2* R614* mutation (*Figure 6D*). Genes upregulated in CMK-*STAG2-/-* markedly increased the number of Wnt-sensitive biological pathways following WNT3A treatment compared with parental CMK cells (*Figure 6E*). A strongly upregulated cluster of 244 transcripts in CMK-*STAG2-/-* included genes encoding signaling molecules (JAG2, IL6ST, SEMA3G) and transcription factors (SP7, KLF3, SMAD3, EP300, and EPHA4) (*Figure 6—figure supplement 2D*). Genes in this cluster were enriched for LEF1 and TCF7 binding sites, indicating their potential to be directly regulated by β-catenin. Enriched biological pathways included Wnt signaling, cell cycle and metabolism (*Figure 6—figure supplement 2E*). Pathway analyses of genes significantly downregulated only in CMK-*STAG2-/-* also showed enrichment for metabolism (*Figure 6—figure supplement 2F,G*).

Overall, our results show that CMK-*STAG2-/-* cells are exquisitely sensitive to Wnt signaling. Introduction of the *STAG2* R614* mutation amplified expression of Wnt-responsive genes and sensitized

genes and pathways that are not normally Wnt-responsive in CMK. This sensitivity could be due at least in part to stabilized β-catenin.

## Conservation of WNT sensitivity in cohesin-deficient zebrafish

To determine if enhanced Wnt sensitivity is conserved in a cohesin-deficient animal model, we used two previously described zebrafish cohesin mutants: stag2b[nz207] (*Ketharnathan et al., 2020*), which has a seven base-pair deletion in stag2b leading to a prematurely truncated Stag2b protein, and rad21[nz171] (*Horsfield et al., 2007*), which has a nonsense point mutation in the rad21 gene that eliminates Rad21 protein. To provide a readout for Wnt signaling, we used transgenic zebrafish carrying a TCF/β-catenin reporter in which exogenous Wnt stimulation induces nuclear mCherry: *Tg(7xTCF-Xla.Siam:nslmCherry)[ia5]* (*Moro et al., 2012*). The Wnt reporter construct, which drives nuclear mCherry red fluorescence in Wnt-responsive cells, was introduced into zebrafish carrying the stag2b[nz207] and rad21[nz171] mutation by crossing.

Zebrafish embryos heterozygous for *Tg(7xTCF-Xla.Siam:nslmCherry)[ia5]*, and either homozygous for stag2b[nz207], rad21[nz171], or wild type, were exposed to 0.15 M LiCl, an agonist of the Wnt signaling pathway (*Figure 7*). Expression of mCherry in the midbrain of embryos was detected at 20 hr post-fertilization (hpf) by epifluorescence and confocal imaging. A constitutive low level of mCherry is present in the developing midbrain of both untreated wild type (*Figure 7A,B,I,J*), and rad21[nz171] embryos (*Figure 7M,N*). On the other hand, untreated stag2b[nz207] embryos exhibited noticeably higher basal mCherry levels than wild type siblings (*Figure 7E,F* compared with A,B). This observation indicates that the Wnt pathway is more intrinsically active in these embryos. However, the addition of LiCl did not result in much additional fluorescence owing to stag2b[nz207] mutation (*Figure 7G, H* compared with C,D). While mCherry expression in rad21[nz171] mutants resembled that in wild type embryos at baseline, LiCl treatment dramatically increased the existing midbrain mCherry expression in rad21[nz171] mutants compared with wild type embryos (*Figure 7O,P* compared with K,L). This observation indicates that rad21[nz171] mutants are more sensitive to Wnt stimulation than wild type.

Overall, the results show that Wnt signaling is sensitized in cohesin-mutant zebrafish embryos, similar to our observations in MCF10A and CMK cell lines. Our findings agree with previous work showing that canonical Wnt signaling is hyperactivated in cohesin-loader *nipblb*-loss-of-function zebrafish embryos (*Mazzola et al., 2019*). Altogether, the results suggest that hyperactivation of Wnt signaling is a conserved feature of cohesin-deficient cells, and that enhanced sensitivity to Wnt is at least in part due to stabilization of β-catenin.

## Discussion

The cohesin complex and its regulators are encoded by several different loci, and genetic alterations in any one of them may occur in up to 26% of patients included in The Cancer Genome Atlas (TCGA) studies (*Romero-Pérez et al., 2019*). Therefore, we were motivated to identify compounds that interfere with cell viability in more than just one type of cohesin mutant. The generation of *RAD21*, *SMC3,* and *STAG2* cohesin mutations in the breast epithelial cell line MCF10A resulted in mild cell cycle and nucleolar phenotypes that are consistent with the many cellular roles of cohesin. Differences between *RAD21* and *SMC3* heterozygotes vs *STAG2* homozygotes could be explained by the fact that RAD21 and SMC3 are obligate members of the cohesin ring, whereas STAG2 can be compensated by STAG1.

Synthetic lethal sensitivities of cohesin-mutant cells that emerged from our screen mostly related to the phenotypes of cohesin-mutant cells, and/or their previously identified vulnerabilities. For example, cohesin-mutant cells were sensitive to inhibitors of the PI3K/Akt/mTOR pathway that feeds into ribosome biogenesis, epigenetic inhibitors that could interfere with cohesin's genome organization and gene expression roles, and a limited sensitivity to PARP inhibitors. The observed sensitivity to PI3K/AKT/mTOR inhibitors is consistent with the nucleolar disruption present in cohesin-deficient cell lines, and with cohesin's known involvement in rDNA transcription and ribosome biogenesis (*Bose et al., 2012*). We have also previously described evidence for sensitivity of cohesin-deficient cell lines to bromodomain (BET) inhibition (*Antony et al., 2020*). However, the Wnt agonist LY2090314, which mimics Wnt activation by inhibiting GSK3-β (*Atkinson et al., 2015*), emerged as a novel class of compound that inhibited the growth of all three cohesin mutants tested. Because there is pathway convergence between Wnt signaling and the PI3K/AKT/mTOR pathway

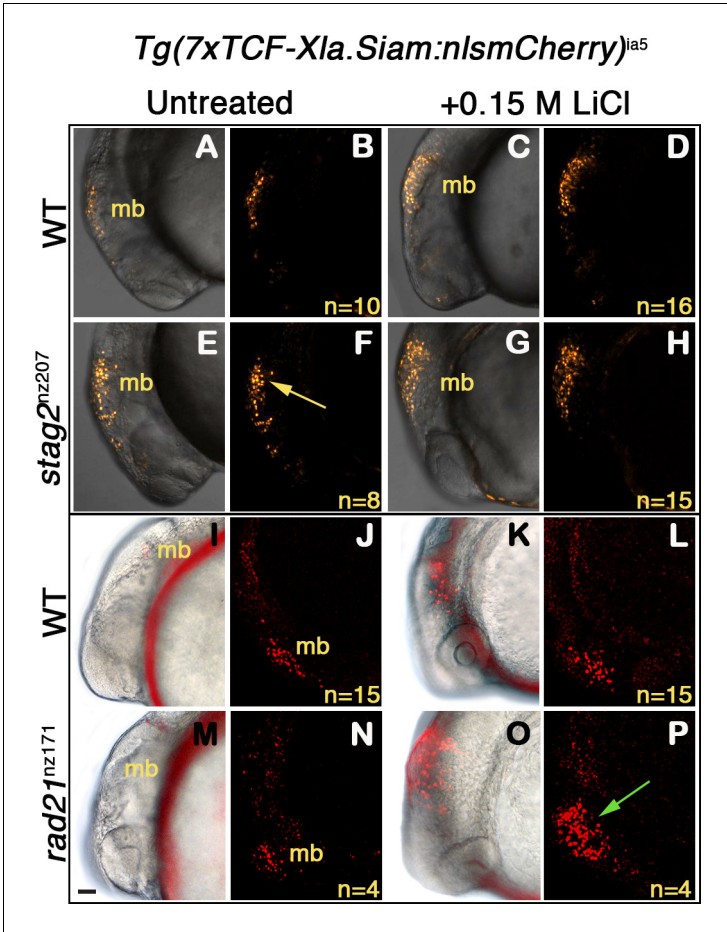

**Figure 7.** Zebrafish *stag2b* and *rad21* cohesin mutants show increased sensitivity to Wnt signaling. Wnt reporter *Tg(7xTCF-Xla.Siam:nlsmCherry)*[ia5] control embryos, *Tg(7xTCF-Xla.Siam:nlsmCherry)*[ia5];*stag2b*[nz207] and *Tg(7xTCF-Xla.Siam:nlsmCherry)*[ia5];*rad21*[nz171] cohesin-mutant embryos were treated with 0.15 M LiCl from 4 hpf to 20 hpf. (**A–H**) max projections of 4 (10 μm) optical sections. (**A,C,E,G**) TD (transmitted light detector) images merged with confocal images. (**B,D,F,H**) confocal images alone. (**I,K,M,O**) Brightfield/fluorescent and (**J,N,L,P**) confocal images of the same embryos in I,K,M,O. (**A–D**) and (**I–J**) *Tg(7xTCF-Xla.Siam:nlsmCherry)*[ia5] control embryos (WT) have low level fluorescence (Wnt reporter activity) in the midbrain that is increased following treatment. (**E–H**) *Tg(7xTCF-Xla.Siam:nlsmCherry)*[ia5]; *stag2b*[nz207] embryos have elevated baseline levels of fluorescence (Wnt reporter activity) relative to controls (yellow arrow) with not much further increase upon LiCl treatment. (**M–P**) *Tg(7xTCF-Xla.Siam:nlsmCherry)*[ia5]; *rad21*[nz171] cohesin-mutant embryos show enhanced Wnt reporter activity in the midbrain (green arrow) upon LiCl treatment, relative to controls. hpf, hours post-fertilization. mb, midbrain. Scale bar, 50 μm.

(*Prossomariti et al., 2020*) it is possible that these pathways represent a common avenue of sensitivity. The identification of PI3K/AKT/mTOR inhibitors in the screen supports this notion.

Wnt signals are important for stem cell maintenance and renewal in multiple mammalian tissues (*Clevers et al., 2014*). Canonical Wnt signaling is required for self-renewal of leukemia stem cells (LSCs) (*Kang et al., 2020*), and is reactivated in more differentiated granulocyte-macrophage progenitors (GMPs) when they give rise to LSCs (*Wang et al., 2010*). Wnt signaling is an important regulator of hematopoietic stem cell (HSC) self-renewal as well (*Rattis et al., 2004*). Our results suggest that cohesin-deficient cells are sensitive to Wnt agonism because they already have stabilization of β-catenin. Interestingly, multiple experimental systems showed that dose-dependent reduction in cohesin function causes HSC expansion accompanied by a block in differentiation (*Viny and Levine, 2018*; *Mazumdar and Majeti, 2017*). It is possible that the effects of cohesin deficiency on HSC development could be in part due to enhanced Wnt signaling.

Cohesin mutations are particularly frequent (53%) in Down Syndrome-associated Acute Megakaryoblastic Leukemia (DS-AMKL) patients (*Yoshida et al., 2013*). In the edited DS-AMKL cell line

*STAG2*-CMK, there was a dramatically enhanced immediate early transcriptional response to WNT3A, including induction of hundreds of genes that are not normally responsive to WNT3A in this cell line. It is possible that STAG2 deficiency in CMK leads to an altered chromatin structure that sensitizes genes to Wnt signaling. However, we observed increased nuclear localization of β-catenin in *STAG2*-CMK, increasing the likelihood that stabilization of β-catenin in *STAG2*-CMK cells accounts at least in part for amplified gene expression in response to WNT3A. Interestingly, DS-AMKL leukemias are often associated with amplified Wnt signaling caused by Trisomy 21 (*Emmrich et al., 2014*) suggesting a possible synergy between cohesin mutations and Wnt signaling dysregulation in DS-AMKL.

We observed an enhanced Wnt reporter response to Wnt activation in *rad21*- and *stag2b*-mutant zebrafish, arguing that Wnt sensitivity upon cohesin mutation is a conserved phenomenon. Previous research shows that cohesin genes are at once both targets (*Xu et al., 2014*; *Ghiselli et al., 2003*) and upstream regulators (*Estarás et al., 2015*; *Schuster et al., 2015*; *Avagliano et al., 2017*; *Mazzola et al., 2019*) of Wnt signaling. For example, depletion of cohesin-loader *nipbl* in zebrafish embryos was reported to downregulate Wnt signaling at 24 hpf (*Pistocchi et al., 2013*), but to upregulate it at 48 hpf (*Mazzola et al., 2019*). What determines the directionality of cohesin's effect on Wnt signaling is unclear. It is possible that feedback loops operate differently in cell- and signal-dependent contexts (*MacDonald et al., 2009*) in cohesin-deficient backgrounds.

Is Wnt pathway activation a conserved feature of cohesin-mutant cancers? Using publicly available data at TCGA (*Grossman et al., 2016*), we correlated expression of genes in the Wnt pathway (Hallmark WNT β-catenin signaling) with nonsense mutation of *STAG2*, the most common of the cohesin mutations, in the four most represented cancers (bladder cancer, endometrial carcinoma, glioblastoma multiforme and cervical kidney renal papillary cell carcinoma). In comparative ranking, the Wnt pathway ranks 8[th] for enrichment in upregulated genes (after DNA repair/G2 checkpoint, MYC/E2F targets, oxidative phosphorylation and the unfolded protein response; *Supplementary file 2*). While our analysis could indicate conserved upregulation of the Wnt pathway in *STAG2* mutant cancers, a caveat is that the true situation is likely to be more complex. If β-catenin is stabilized, upstream and in-parallel effectors such as Wnt ligands, receptors, signaling intermediates might trend to downregulation owing to negative feedback loop regulation of the Wnt pathway for example via DKK1 (*Niida et al., 2004*). Furthermore, because cancer collections are unlikely to be under Wnt stimulation at the time of RNA collection, it is possible that their true Wnt sensitivity will be missed in a steady-state transcription analysis.

A remaining question is exactly what causes β-catenin stabilization and Wnt hyperactivation in cohesin-mutant cells. One possibility is that stabilization of β-catenin upon cohesin deficiency is linked to energy metabolism. For example, in developing amniote embryos, glycolysis in actively growing cells of the embryonic tail bud raises the intracellular pH, which in turn causes β-catenin stabilization (*Oginuma et al., 2020*; *Oginuma et al., 2017*). This mode of glycolysis is similar to that observed in cancer cells and is known as the Warburg effect (*Parks et al., 2013*; *Wang et al., 2016*). Metabolic and oxidative stress disturbances in cohesin-mutant cells observed in this study and others (*Bose et al., 2012*; *Cukrov et al., 2018*) might similarly influence β-catenin stability and in turn, sensitivity to incoming Wnt signals. Further research will be needed to determine how β-catenin is stabilized in cohesin-deficient cells, whether β-catenin stabilization is part of a transition to malignancy, and if the associated Wnt sensitivity represents a therapeutic opportunity in cohesin-mutant cancers.

## Materials and methods

### Cell culture

MCF10A (a spontaneously immortalized breast epithelial cell line) was purchased from Sigma (product #: CRL 10317). Sigma use STR analysis to verify the line. A PCR test in our laboratory showed that all cells were mycoplasma free. CMK (acute megakaryocytic leukemia associated with Down Syndrome) was a gift from Dr Motomi Osato, National University of Singapore. These cells tested negative with MycoAlert Mycoplasma Detection Kit in our laboratory. K562 (chronic myelogenous leukemia) were purchased from ATCC, CCL-243. These cells tested mycoplasma free using B3903-Mycoplasma Detection Kit - QuickTest-com from Biotool. The HTC116 cell line was purchased form ATCC. STR analysis is used by ATCC to verify all cell lines.

MCF10A and its cohesin-deficient derivatives were maintained in Dulbecco's modified Eagle medium (DMEM) (Life Technologies) supplemented with 5% horse serum (Life Technologies), 20 ng/mL of epidermal growth factor (EGF) (Peprotech), 0.5 mg/mL of hydrocortisone, 100 ng/mL of cholera toxin and 10 µg/mL of insulin (local pharmacy). All supplements were purchased from Sigma-Aldrich unless otherwise stated. K562 cells were maintained in Isocove's Modified Dulbecco's Media (IMDM) (Life Technologies) containing 10% fetal bovine serum. CMK cells were maintained in RPMI 1640 media containing 10% fetal bovine serum. CMK cells and adherent K562-*STAG2* null line were detached for subculture using trypsin-EDTA (0.005% final concentration, Life Technologies). For WNT stimulation, recombinant Human WNT-3A Protein (5036-WN, R and D systems) was used at 200 ng/mL for 4 hr. Human colorectal carcinoma HCT116 cells were grown in DMEM with 10% fetal calf serum and antibiotics. Stable transfection of HCT116 cells with *SMC1A* mutations c.2027A > G and c.2479 C > T was described previously (*Sarogni et al., 2019*). All cells were cultured at 37°C in 5% $CO_2$.

## Generation of isogenic cell lines using CRISPR-CAS9 editing

We used CRISPR-CAS9 sgRNAs targeting the 5' and 3' UTR regions of *RAD21*, *SMC3* and *STAG2* gene to create MCF10A *RAD21+/-*, MCF10A *SMC3+/-*, and MCF10A *STAG2-/-*. Specific sgRNAs were cloned into the px458 plasmid (*Ran et al., 2013*) and transfected into MCF10A cells. Single GFP-positive cells were isolated into 96-well plates using a FACSAriaII (Becton Dickinson) and clonally expanded. PCR screening and Sanger sequencing identified cells with heterozygous deletion of *RAD21* or *SMC3* and homozygous deletion of *STAG2*. Primer and guide sequences are provided in *Supplementary file 1*. Editing of K562 to create STAG2 R614* mutation has been described previously (*Antony et al., 2020*). The CMK line with the STAG2 R614* mutation was generated using the same sgRNA and method.

## Quantitative PCR (RT-qPCR)

Total RNA was extracted using NucleoSpin RNA kit (Machery-Nagel). cDNA was synthesized using qScript cDNA SuperMix (Quanta Biosciences). RT-qPCR was performed on a LightCycler 480 II (Roche Life Science) using SYBR Premix Ex Taq (Takara). Expression values relative to reference genes *cyclophilin* and *glyceraldehyde 3-phosphate dehydrogenase* (*GAPDH*) were derived using qBase Plus (Biogazelle). Primer sequences are provided in *Supplementary file 1*.

## Antibodies

Primary antibodies used are as follows: anti-RAD21 (ab992), anti-SMC3 (#5696, CST), anti-STAG2 (ab4463), anti-γ-Tubulin (T5326; Sigma-Aldrich) in 1:5000, anti-fibrillarin (ab5821), anti-nucleolin (ab13541), anti-gamma H2AX (ab26350), anti-TP53 (ab131442), anti-β-catenin (#9562, CST), anti-phospho-β-catenin (Ser675) (#9567, CST), anti-phospho- phospho-β-catenin (Ser33/37/Thr41) (#9561, CST). Secondary antibodies used for immunofluorescence were anti-mouse Alexa Fluor 488 (1:2000, Life Technologies), anti-rabbit Alexa Fluor 488 (1:2000, Life Technologies), anti-rabbit Alexa Fluor 568 (1:2000, Life Technologies). All antibodies were used in 1:1000 for immunoblotting or immunofluorescence unless otherwise stated.

## Immunoblotting

Immunoblotting was performed as described previously (*Antony et al., 2015*; *Dasgupta et al., 2016*). Primary antibodies were detected using IRDye 800CW Donkey anti-Goat IgG and IRDye 680CW Goat anti-mouse IgG, IRDye 800CW Goat anti-rabbit IgG (LICOR). LI-COR Odyssey and LI-COR Image Studio software was used to image and quantify blots.

## Proliferation assays

MCF10A parental and isogenic cohesin-deficient cell lines were seeded in 96-well plates at 2000 cells per well. Cell confluence was monitored by time-lapse microscopy using IncuCyte FLR with a 10X objective for 5 days.

## Cell cycle analysis

Cells were synchronized using double thymidine block as described previously (*Dasgupta et al., 2016*). Samples were harvested, fixed, and stained with 10 µg/mL propidium iodide (PI) and 250 µg/mL RNase A, 37°C for 30 min. Cells were then analyzed using a Beckman Coulter Gallios Flow Cytometer.

## Immunofluorescence and imaging

Cells stained for FBL, gH2AX, and TP53 were imaged using an Opera Phenix high content screening system, with 63x water objectives in confocal mode. Spot enumeration and signal intensities were analyzed using Harmony software (PerkinElmer). CMK or MCF10A cells were fixed with 4% (v/v) paraformaldehyde in PBS for 10 min, then permeablized with 0.1% Triton in PBS. CMK cells were spun onto slides using SHANDON cytospin prior to fixation. Cells were blocked with 2% (w/v) bovine serum albumin in PBS, then incubated with primary antibody overnight at 4°C. The next day, cells were washed with PBS and incubated with secondary antibody and Hoechst 33342 (1 µg/mL) at room temperature for 1 hr. Cells were then washed and mounted on slides using ProLong Gold Anti-fade (Life Technologies) or DAKO mounting medium. Imaging was done using a Nikon C2 confocal microscope with NIS Elements software and images were processed and quantified using Image J or FIJI software.

## High-throughput compound screen

3009 compounds from an FDA-approved, kinase inhibitors and epigenetic libraries (Compounds Australia, Griffith University, Australia) were screened against MCF10A parental and MCF10A cohesin-deficient clones. Compounds are stored by Compounds Australia under robust environmental conditions and supplied in assay ready plate format. MCF10A parental cells and cohesin-deficient derivatives were screened as single biological replicates with two technical replicates for each drug concentration. MCF10A parental cells were seeded at 600 cells/well, *RAD21+/-* at 800 cells/well, *SMC3+/-* at 1200 cells/well and *STAG2-/-* at 700 cells/well, into 384-well CellCarrier-384 Ultra microplates (PerkinElmer) with EL406 microplate washer dispenser (BioTek). 24 hr later, growth media was aspirated from the plates and 35 µL of fresh MCF10A complete growth media was added into each well using BioTek EL406TM Microplate washer dispenser. 5 µL of 8x concentrated compound solution (diluted in MCF10A complete growth media) was added into each well with a JANUS automated liquid handling system (PerkinElmer). At the time of drug treatment, one untreated plate was retained to determine the cell number at t = 0. For the rest of the assay plate, compounds were added in 40 µL of complete medium per well using an Echo 550 Liquid Handler (Labcyte). Positive (Campthothecin, 40 nM) and negative controls (DMSO 0.1%) were added to each assay plate for quality control. Duplicated control plates of each cell line were also prepared for cell count quantification prior to the addition of drugs, to be used in the growth rate inhibition (GR) metrics. After 48 hr, control and treated cells in were fixed and stained simultaneously using 4% PFA/0.5 µg/mL DAPI/0.1% Triton X-100 solution. Cells were imaged using an Opera Phenix high content screening system and analyzed using Columbus software (PerkinElmer) using 20x water objectives. To determine compound activity, dose-response curves for each compound was plotted and using an R package, GRmetrics (*Clark et al., 2017*). Synthetic lethal candidate compounds were selected and ranked based on the differential area over curve (AOC) metrics (*Hafner et al., 2016*) derived from dose-response curves of MCF10A parental and cohesin-deficient cell lines. Compounds that caused ≥30% growth inhibition (AOC ≥0.15) in cohesin-deficient clones compared with parental MCF10A cells were selected as hits. A secondary screen was performed with 85 candidate hits identified from primary screen (two technical replicates per clone per concentration). The secondary screen was added to independently validate primary screen data. Activity of the selected compounds was tested in eleven concentrations (0.5 nM to 10 µM). Secondary screen hit compounds were selected based on the same threshold used in primary screen.

## Cell viability assays to validate compounds identified from the screen

Individual compounds were purchased from Sigma-Aldrich, Selleck Chem or MedChemExpress and dissolved in DMSO at recommended concentrations. MCF10A parental cells were seeded in 96-well plates at 3000 cells/well, *RAD21+/-* at 4500 cells/well, *SMC3+/-* at 5000 cells/well and *STAG2-/-* at

3500 cells/well. Cells were incubated for 24 hr then treated with compounds for 48 hr. DAPI-stained cells were counted using a Lionheart FX automated microscope (BioTek). K562 parental and *STAG2-/-* cells were seeded at 2000 cells per well in 96 well plates, incubated with the compound for 48 hr after which viability was measured using 3-(4,5-dimethylthiazol-2-yl)−2,5-diphenyltetrazolium bromide or MTT.

## RNA-sequencing and analyses

Total RNA was extracted using NucleoSpin RNA kit (Machery-Nagel). MCF10A libraries from three biological replicates of each cell type were prepared using NEBNext Ultra RNA Library Prep Kit (Illumina) and sequenced on HiSeq X by Annoroad Gene Technology Ltd. (Beijing, China), contracted through Custom Science (NZ). CMK lines were treated with 200 ng/mL of recombinant human WNT3A (R and D systems) for 4 hr. Libraries were prepared from baseline (non-treated) and WNT3A treated CMK cell lines using Illumina TruSeq stranded mRNA library and sequenced on the HiSeq 2500 V4 at the Otago Genomics Facility (Dunedin, New Zealand). RNA-sequencing reads were first trimmed for sequencing adapters and low quality (q < 20). Cleaned reads were then aligned to the human genome GRCh37 (hg19) using HISAT2 version 2.0.5 with gene annotation from Ensembl version 75. Read counts were retrieved by exon and summarized by gene using featureCount (*Liao et al., 2014*) version v1.5.3. Differentially expressed genes in the *STAG2-/-* mutants versus wild type were identified using DESeq2 (*Love et al., 2014*). P-values were adjusted for multi-test following Benjamini-Hochberg methodology. Pathways analyses were performed using Molecular Signature Databases ReactomePA (*Yu and He, 2016*) and clusterProfiler (*Yu et al., 2012*) on differentially expressed genes.

## Zebrafish methods and imaging

Wild type (WIK), TCF reporter line *Tg(7xTCF-Xla.Siam:nslmCherry)*[ia5] (*Moro et al., 2012*), *stag2b*[nz207] (*Ketharnathan et al., 2020*) and *rad21*[nz171] (*Horsfield et al., 2007*) mutant fish lines were maintained according to established husbandry methods (*Westerfield, 1995*). We crossed heterozygous *rad21*[nz171] or homozygous *stag2b*[nz207] mutants to the TCF reporter line then crossed these fish (*rad21*[nz171]/+;TCF/+) to *rad21*[nz171]/+, or (*stag2b*[nz207]/+;TCF/+) to *stag2b*[nz207]/+. Progeny carrying the TCF reporter were incubated with 0.15 M lithium chloride (LiCl) from 4 hpf for 16 hr. LiCl salt was directly dissolved in embryo water and added to embryos in six well plates. 50 embryos were used per treatment group. At 20 hpf, embryos were washed, anesthetized with MS-222 (200 mg/L) and mounted in low melting agarose (0.6%) or 3% methyl cellulose for imaging. Z-stacks were acquired using a Nikon C2 confocal microscope. Embryos were genotyped post-imaging.

## Statistical analyses

All statistical analyses were carried out using R Studio or Prism version eight software (GraphPad).

# Acknowledgements

The authors are grateful to the ANU Centre for Therapeutic Discovery at the John Curtin School of Medical Research at Australian National University for assistance with screening. We acknowledge Compounds Australia (https://www.compoundsaustralia.com) for their provision of specialized compound management and logistics research services to the project.

# Additional information

## Funding

| Funder | Grant reference number | Author |
| --- | --- | --- |
| Health Research Council of New Zealand | 15/229 | Julia A Horsfield |
| Health Research Council of New Zealand | 19/415 | Ross D Hannan<br>Julia A Horsfield |
| Associazione Italiana per la | IG23284 | Antonio Musio |

Ricerca sul Cancro

| The Maurice Wilkins centre for Molecular Biodiscovery | 3705733 | Jisha Antony Julia A Horsfield |

The funders had no role in study design, data collection and interpretation, or the decision to submit the work for publication.

### Author contributions

Chue Vin Chin, Sarada Ketharnathan, Conceptualization, Formal analysis, Investigation, Methodology, Writing - original draft, Writing - review and editing; Jisha Antony, Conceptualization, Formal analysis, Supervision, Funding acquisition, Investigation, Methodology, Writing - original draft, Writing - review and editing; Anastasia Labudina, Maria Michela Pallotta, Formal analysis, Investigation; Gregory Gimenez, Data curation, Formal analysis, Visualization; Kate M Parsons, Investigation, Methodology; Jinshu He, Investigation, Methodology, Writing - review and editing; Amee J George, Conceptualization, Formal analysis, Methodology, Writing - review and editing; Antonio Musio, Formal analysis, Supervision, Funding acquisition; Antony Braithwaite, Parry Guilford, Conceptualization, Supervision, Methodology, Writing - review and editing; Ross D Hannan, Conceptualization, Resources, Supervision, Methodology; Julia A Horsfield, Conceptualization, Formal analysis, Supervision, Funding acquisition, Investigation, Methodology, Writing - original draft, Project administration, Writing - review and editing

### Author ORCIDs

Amee J George (iD) http://orcid.org/0000-0002-0265-4476
Antonio Musio (iD) http://orcid.org/0000-0001-7701-6543
Julia A Horsfield (iD) https://orcid.org/0000-0002-9536-7790

### Ethics

Animal experimentation: Work with zebrafish was approved by the University of Otago (Dunedin) Animal Ethics Committee (AUP19/17) and conducted using approved institutional animal care standard operating procedures.

### Decision letter and Author response

Decision letter https://doi.org/10.7554/eLife.61405.sa1
Author response https://doi.org/10.7554/eLife.61405.sa2

## Additional files

### Supplementary files

• Supplementary file 1. List of sgRNA sequences and PCR primers.

• Supplementary file 2. TCGA analysis of STAG2 mutant vs wild type cancers.

• Transparent reporting form

### Data availability

All RNA sequencing data has been deposited at the GEO database under accession codes GSE154086. All data generated or analysed during this study are included in the manuscript and supporting files. Source data files have been provided for Figures 1-5 and Table 1.

The following dataset was generated:

| Author(s) | Year | Dataset title | Dataset URL | Database and Identifier |
|---|---|---|---|---|
| Chin CV, Antony J, Gimenez G, Horsfield J | 2020 | Expression profiling in cohesin mutant MCF10A epithelial and CMK leukaemia cells | https://www.ncbi.nlm.nih.gov/geo/query/acc.cgi?acc=GSE154086 | NCBI Gene Expression Omnibus, GSE154086 |

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
