## [Decision Letter]

**Acceptance summary:**

The manuscript describes an interesting finding that stimulation of Wnt signaling can lead to lethality in cohesin mutant cells. This suggests a potentially novel therapeutic strategy for cohesion-mutant cancers.

**Decision letter after peer review:**

Thank you for submitting your article "Cohesin mutations are synthetic lethal with stimulation of WNT signaling" for consideration by *eLife*. Your article has been reviewed by three peer reviewers, one of whom is a member of our Board of Reviewing Editors, and the evaluation has been overseen by Richard White as the Senior Editor. The following individuals involved in review of your submission have agreed to reveal their identity: Aaron Viny (Reviewer #2); Massa Valentina (Reviewer #3).

The reviewers have discussed the reviews with one another and the Reviewing Editor has drafted this decision to help you prepare a revised submission.

Summary:

In this manuscript, Chin et al., investigate synthetic lethal candidates in conjunction with cohesin mutations in cancer. The authors engineer mutant cell line variants for several members of the cohesin complex and demonstrate that they exhibit established features of cohesin-mutant cells. Then they conduct a small molecule screen to identify compounds and pathways with increased sensitivity in these cohesin-mutant cells. They identify a GSK3 inhibitor suggesting that Wnt pathway activation is synthetic lethal with cohesin mutants. They show evidence suggesting this occurs though β-catenin stabilization and activation of Wnt target genes. Finally, they use a genetic zebrafish model to show hyperactivation of Wnt pathway in Rad21-deficient embryos. In general, the link between cohesin mutants and Wnt pathway activation is of interest, but there are several issues with the manuscript.

Essential revisions:

1) While there is evidence presented here that cohesin-mutant cells exhibit increased Wnt pathway activation, there is no data demonstrating that it is key to the cancer phenotypes of cohesin-mutant cancers. Such data would enhance the significance of the findings here.

2) Analysis of the cohesin-mutant MCF10A clones relied on investigation of a single clone per gene. This is not sufficient to draw convincing conclusions, particularly with CRIPSR/Cas9 targeting and potential off target effects. Several clones per gene should be investigated to account for potential clonal variability, particularly given the differences in phenotypes observed between these clones.

3) The chemical screen identified a number of cohesin mutant sensitivities, and the common hits are shown, however, given the phenotypic differences between the RAD21/SMC3 clone and the STAG2 null clones in micronuclei and lagging strands, did the 76 STAG2 unique chemical compounds share a functional class? Why do the authors think SMC3 and STAG2 had far more overlap that each of these did with RAD21? Why did RAD21 have so few hits overall by comparison?

4) Western blotting shows cohesin mutant clones have increased total and membrane β-catenin that lead to transcriptional responses. Is there evidence of Wnt pathway/β-catenin transcriptional activation in publicly available datasets of cohesin perturbation or cohesin-mutant cancers?

5) The zebrafish experiments are fundamental and elegantly done. However, I think that, given the role of RAD21 and of the other cohesin subunits, it would be pivotal to include similar experiments with zebrafish modelling other genes of the complex.

6) Cohesin subunit deficient clones of MCF10A were generated using CRISPR-Cas9 editing. While this is clearly described in the Materials and methods section, it would be helpful to state the model system as well in results.

7) For cell cycle analysis (Results section) it would be extremely important, also considering the literature, to assess cell death rate.

8) In Figure 1F/1G/Figure 1—figure supplement 1B what do the square symbols represent? Please include in the legend.

9) In Figure 4A/B/Figure 4—figure supplement 1A/1B please include size markers. Also Figure 4—figure supplement 1 needs a loading control such as tubulin.

---

## [Author Response]

Essential revisions:1) While there is evidence presented here that cohesin-mutant cells exhibit increased Wnt pathway activation, there is no data demonstrating that it is key to the cancer phenotypes of cohesin-mutant cancers. Such data would enhance the significance of the findings here.

The reviewers make a good point. It would be ideal to test if cohesin mutant primary cancer cells are Wnt-sensitivity or hyperactivity as part of their phenotype, but actually testing this in vivo is quite a difficult task. Primary cells with and without cohesin mutations would need to be compared in a Wnt-stimulated scenario, because there may be no apparent phenotype to measure in cells at baseline or endpoint conditions where Wnt signalling is not active.

While we unfortunately do not have access to primary cells, in a revision experiment for this paper we managed to show that Wnt sensitivity is extendable to another cancer cell type (colorectal cancer cell line HCT116), and a different cohesin subunit mutation: *SMC1A*.

To provide the new data, we collaborated with our Italian colleague Antonio Musio, whose lab has the appropriate models. Antonio’s group have characterised two cohesin SMC1A mutations identified in human colorectal carcinomas, c.2027A>G and c.2479 C>T (Cucco et al., 2014; Sarogni et al., 2019). Using the same treatment conditions as we did for MCF10A cells, Antonio’s group showed increased b-catenin protein levels, particularly in response to LY2090314, in HCT116 cells that stably overexpress SMC1A containing these two different mutations. Importantly, SMC1A is a cohesin gene that wasn’t tested in our screen or subjected to downstream analysis in our lab, and yet these experiments confirm our independent findings.

We have added a new figure supplement with these data (Figure 5—figure supplement 2). Antonio and colleague Michela Pallotta (who performed the work) have been included as co-authors on the manuscript. The new text in the results (which also explains the model’s limitation) is reproduced below:

“To determine if stabilization of β-catenin is conserved in a second model of cohesin mutant cancer, we performed an identical LY2090314 treatment on HCT116 cells that were stably transfected to express two SMC1A mutations identified in human colorectal carcinomas, c.2027A>G (leading to p.E676G change near the hinge domain) and c.2479 C>T (leading to a truncated protein, p.Q827X) [53, 54]. The limitation is that these cells are not a model of cohesin deficiency, but rather, one in which normal cohesin function is perturbed by expression of a mutant version of SMC1A [54]. We observed an increased basal level of phosphorylated β-catenin at Ser675 in cells that express either of these SMC1A mutants. Furthermore, LY2090314 treatment markedly increased total β-catenin in both the SMC1A mutants compared with HCT116 wild type controls (Figure 5—figure supplement 2A,B). The results indicate that abnormally high levels of active β-catenin are also present following Wnt stimulation when a fourth subunit of cohesin, SMC1A, is perturbed.”

Interestingly, the mutant SMC1A-expressing HCT116 cells are more aggressive in tumour formation (Sarogni et al., 2019). Future experiments beyond the scope of this study could follow up to determine if Wnt signalling is important for this phenotype.

Below in response to point #4, we describe an additional analysis of TCGA data, which supports the idea that cohesin mutant cancers have over-active Wnt signalling.

2) Analysis of the cohesin-mutant MCF10A clones relied on investigation of a single clone per gene. This is not sufficient to draw convincing conclusions, particularly with CRIPSR/Cas9 targeting and potential off target effects. Several clones per gene should be investigated to account for potential clonal variability, particularly given the differences in phenotypes observed between these clones.

Owing to the expense of high throughput screening, it was too costly to screen more than one clone per gene in the screen. However, even though there’s only a single clone representing each cohesin subunit in the SL screen, we believe the screen was actually quite conservative and robust, because the top hits we followed up were only selected if they affected all three cohesin mutant clones.

However, in response to this valid comment, we have now analysed one additional RAD21 and one addition SMC3 clone that were produced, but not used in the screen. These clones (RAD21+/- clone 2 and SMC3 +/- clone 2) behave similarly to the original clones that were used in the screen. Data showing allele details, cell morphology, mRNA levels, chromosome cohesion defects, cell viability, growth curves and response to LY2090314 are shown in Figure 3—figure supplement 3. The cells grow slightly more slowly than wild type and are differentially sensitive to LY2090314. These results indicate that the clones used for the screen were likely to be typical and any variability was not likely to be due to off-target effects. The new data are described in the Results section:

“Collectively, characterization of our cohesin-deficient clones provided confidence that they represent suitable models for synthetic lethal screening. To confirm that the phenotypes of our chosen clones are representative, we selected a further two clones with heterozygous deletions in SMC3 and RAD21, and monitored their growth, morphology and chromosome cohesion (Figure 3—figure supplement 3). These analyses showed that the two additional deletion clones were similar to those already characterized (Figure 1, Figure 2, Figure 3), providing surety that our cohesin-deficient cell lines have properly representative phenotypes. Furthermore, none of the cohesin-deficient clones exhibited enhanced cell death compared with parental MCF10A cells (Figure 2B; Figure 3—figure supplement 3).”

Regarding the subtly different phenotypes of the cohesin-deficient clones, it is perhaps not surprising that the clones do not have identical phenotypes, particularly for the *STAG2* homozygous deletion.

First, STAG2 can be substituted by STAG1 and therefore a *STAG2* mutation can be homozygous, unlike SMC3 and RAD21. This means that in the *STAG2* mutant, STAG1 cohesin is predominant, whereas in *RAD21* and *SMC3* mutants, there would be predicted to be a reduction in dose of STAG1- and STAG2-containing cohesin.

Second, differences in chromosome segregation phenotypes between the different cohesin-deficient clones might be due to STAG2 having a role in centromere chromosome cohesion (Canudas and Smith, 2009) such that its homozygous loss results in “cleaner” mis-segregation events and more frequent complete loss of chromosome cohesion, as shown in Figure 2D (please note that the key of Figure 2D has been corrected to show that the black section of the bar is 'normal' and the 'green' sections correspond to loss of cohesion!).

The functional differences between STAG2 and STAG1 cohesin are described nicely in a recent review from Ana Losada and colleagues (Cuadrado and Losada, 2020).

In the screen there was quite a high degree of overlap of hits for the clones we generated, particularly *SMC3* and *RAD21*, where only 5 of 27 compounds affecting *RAD21* mutants were not also a hit in *SMC3* mutants, and only 1 hit that was exclusive to *RAD21*. This suggests that despite potential differences between the clones, there were a number of common sensitivities. All clones have been well characterised, and findings extended to other cell lines and zebrafish.

3) The chemical screen identified a number of cohesin mutant sensitivities, and the common hits are shown, however, given the phenotypic differences between the RAD21/SMC3 clone and the STAG2 null clones in micronuclei and lagging strands, did the 76 STAG2 unique chemical compounds share a functional class?

The unique hits for each clone are given in Figure 4—figure supplement 2. This figure shows that functional classes are largely shared among the cohesin subunits. From this figure, classes of compound that affected only cells containing the *STAG2* mutation can be identified. These include “cytoskeletal”, “GPCR and G protein”, ‘Immunology and inflammation’ and “transmembrane transporters”. These extra categories specific to *STAG2* mutation look like they reflect a response to particular signalling pathways, and could result from the more specialised role of STAG2-cohesin in tissue-specific transcription.

Why do the authors think SMC3 and STAG2 had far more overlap that each of these did with RAD21?

Figure 4E shows that a number of compounds affecting *RAD21* mutant cells also inhibited *SMC3* mutants (n=4) and *STAG2* mutants (n=4), and there were 18 hits in common with all 3 mutants. In addition, *RAD21* mutant cells were sensitive to only one compound that didn’t also affect the other mutants, and this particular compound is involved in DNA damage repair. It’s just that there were fewer compounds overall that affected *RAD21* mutants. That’s probably why there wasn’t as much overlap as observed with *SMC3* and *STAG2*.

Why did RAD21 have so few hits overall by comparison?

That’s a very good question. We don’t have an answer but can speculate that additional compound sensitivities in *SMC3* and *STAG2* mutants might derive from important dose-dependent functions that they have in the nucleus, which are not equally sensitised by limiting the amount of RAD21-containing cohesin. We wonder if SMC3 depletion exposes an important role in chromosome maintenance because of our observation of increased nuclear decompaction and micronuclei the *SMC3* mutant clone (Figure 2E-G). The specialised role of STAG2 in gene expression (see answer to Q2 above) might give the STAG2 mutant an increased drug sensitivity profile that includes a wider range of pathways.

4) Western blotting shows cohesin mutant clones have increased total and membrane β-catenin that lead to transcriptional responses. Is there evidence of Wnt pathway/β-catenin transcriptional activation in publicly available datasets of cohesin perturbation or cohesin-mutant cancers?

This is an excellent suggestion, we have followed this up bioinformatically using publicly available TCGA data. We could only really look at *STAG2-*mutant cancers with high sample numbers in TCGA, because we needed a decent number of samples that had a cohesin mutation AND gene expression data. Here’s a summary of what we did:

*STAG2* nonsense mutations (frameshift or stop-gained events) with gene expression were retrieved from TCGA by cancer type. Only cancer types having more than two *STAG2* nonsense mutation samples were considered. After this filtering, four cancer types are remaining: Bladder Urothelial Carcinoma (BLCA), Uterine Corpus Endometrial Carcinoma (UCEC), Glioblastoma multiforme (GBM) and Cervical Kidney renal papillary cell carcinoma (KIRP).

For each cancer type, an equal number of control samples were also retrieved. Samples were considered as control if no STAG2 nonsense mutation was present.

When the 150 samples (all cancer type) were ranked by the Wald statistic, STAG2 was the first ranked gene. The comparison of this ranking to the gene set HALLMARK_WNT_Β_CATENIN ranked this pathway as 8th most enriched, overall showing an enrichment in up-regulated genes in the Wnt pathway. The results can be found in Supplementary file 2.

So, in the analysis we did, it looks like Wnt hyperactivation IS a feature of STAG2 mutant cancers. However, we treat the results quite cautiously, and only in the Discussion section, because of the caveats with this approach:

“Is Wnt pathway activation a conserved feature of cohesin-mutant cancers? Using publicly available data at TCGA [74], we correlated expression of genes in the Wnt pathway (Hallmark WNT β-catenin signaling) with nonsense mutation of STAG2, the most common of the cohesin mutations, in the four most represented cancers (bladder cancer, endometrial carcinoma, glioblastoma multiforme and cervical kidney renal papillary cell carcinoma). In comparative ranking, the Wnt pathway ranks 8th for enrichment in upregulated genes (after DNA repair/G2 checkpoint, MYC/E2F targets, oxidative phosphorylation and the unfolded protein response; Supplementary File 2). While our analysis could indicate conserved upregulation of the Wnt pathway in STAG2 mutant cancers, a caveat is that the true situation is likely to be more complex. If β-catenin is stabilized, upstream and in-parallel effectors such as Wnt ligands, receptors, signalling intermediates might trend to downregulation owing to negative feedback loop regulation of the Wnt pathway for example via DKK1 [75]. Furthermore, because cancer collections are unlikely to be under Wnt stimulation at the time of RNA collection, it is possible that their true Wnt sensitivity will be missed in a steady-state transcription analysis.”

5) The zebrafish experiments are fundamental and elegantly done. However, I think that, given the role of RAD21 and of the other cohesin subunits, it would be pivotal to include similar experiments with zebrafish modelling other genes of the complex.

We are happy to include timely new data that addresses this concern.

We recently generated zebrafish carrying mutations for the *stag* genes, *stag1a*, *stag1b* and *stag2b*. Characterisation of the Stag paralogues in zebrafish indicates that Stag2b is the most abundant Stag2, and likely to be most closely related to human STAG2. A description of the mutants and their early haematopoietic phenotypes has recently been accepted for publication (preprint here: https://www.biorxiv.org/content/10.1101/2020.10.19.346122v1).

The *stag2b* mutant is homozygous viable. We crossed this mutant to the TCF reporter line then crossed these fish (*stag2b*/+; TCF/+) to *stag2b*/+. Progeny carrying the TCF reporter were stimulated (or not) with LiCl, imaged, and genotyped post-imaging. Data for (*stag2b*/*stag2b*; TCF/+) and (+/+; TCF/+) genotypes were collected for a revision to Figure 6, now Figure 7 in the manuscript.

The results show higher baseline activity of the TCF Wnt reporter in *stag2b* homozygotes, but when stimulated with LiCl, we could not reliably determine extra expression above WT. Therefore, the *stag2b* mutant zebrafish also show enhanced Wnt signalling, but unlike for *rad21-/-* embryos, the enhancement is at baseline, pre-stimulation. Overall, the zebrafish experiments are consistent with over-active Wnt signalling being associated with cohesin mutation during embryogenesis. This information has been added to the Results:

“Zebrafish embryos heterozygous for Tg(7xTCF-Xla.Siam:nslmCherry)ia5, and either homozygous for stag2bnz207, rad21nz171 or wild type, were exposed to 0.15 M LiCl, an agonist of the Wnt signaling pathway (Figure 7). Expression of mCherry in the midbrain of embryos was detected at 20 hours post-fertilization (hpf) by epifluorescence and confocal imaging. A constitutive low level of mCherry is present in the developing midbrain of both untreated wild type (Figure 7A,B,I,J), and rad21nz171 embryos (Figure 7M,N). On the other hand, untreated stag2bnz207 embryos exhibited noticeably higher basal mCherry levels than wild type siblings (Figure 7E,F compared with A,B). This observation indicates that the Wnt pathway is more intrinsically active in these embryos. However, the addition of LiCl did not result in much additional fluorescence owing to stag2bnz207 mutation (Figure 7G,H compared with C,D). While mCherry expression in rad21nz171 mutants resembled that in wild type embryos at baseline, LiCl treatment dramatically increased the existing midbrain mCherry expression in rad21nz171 mutants compared with wild type embryos (Figure 7O,P compared with K,L). This observation indicates that rad21nz171 mutants are more sensitive to Wnt stimulation than wild type.”

Anastasia Labudina, who performed these experiments, has been added as an author on the manuscript.

6) Cohesin subunit deficient clones of MCF10A were generated using CRISPR-Cas9 editing. While this is clearly described in the Materials and methods section, it would be helpful to state the model system as well in results.

A brief description has been added to subsection “Cohesin gene deletion in MCF10A cells results in minor cell cycle defects”, and Supplementary file 1 has been redesignated as a Figure (Figure 1) describing generation of the mutant clones.

7) For cell cycle analysis (Results section) it would be extremely important, also considering the literature, to assess cell death rate.

We have included trypan blue measurement of cell survival in Figure 3—figure supplement 3E, described in the text: *“*Furthermore, none of the cohesin-deficient clones exhibited enhanced cell death compared with parental MCF10A cells (Figure 2B; Figure 3—figure supplement 3E).”

None of the cohesin-deficient clones have particularly enhanced cell death. Cell survival information is already included in the paper: the cell cycle flow cytometry analyses in Figure 2B do not show a sub-G1 population, which would be indicative of cell death.

8) In Figure 1F/1G/Figure 1—figure supplement 1B what do the square symbols represent? Please include in the legend.

This Figure is now Figure 2. The square symbols refer to the biological replicates for each experiment. There were actually 5 biological replicates per experiment, and this has now been corrected in the figure legend for both figures.

9) In Figure 4A/B/Figure 4—figure supplement 1A/1B please include size markers. Also Figure 4—figure supplement 1 needs a loading control such as tubulin.

This Figure is now Figure 5. For Figure 5A,B, we have provided a file showing the original blots with molecular size markers as Figure 5—source data 1. A loading control and molecular sizing has been added to Figure 5—figure supplement 1A/B. Original blots with molecular sizes have been provided for Figure 5—figure supplement 1 (Figure 5—source data 3) and Figure 5—figure supplement 2 (Figure 5—source data 4).